# Impact of Enniatin B and Beauvericin on Lysosomal Cathepsin B Secretion and Apoptosis Induction

**DOI:** 10.3390/ijms24032030

**Published:** 2023-01-19

**Authors:** Mohammed Aufy, Ramadan F. Abdelaziz, Ahmed M. Hussein, Nermina Topcagic, Hadil Shamroukh, Mostafa A. Abdel-Maksoud, Tamer Z. Salem, Christian R. Studenik

**Affiliations:** 1Department of Pharmaceutical Sciences, Division of Pharmacology and Toxicology, University of Vienna, 1090 Vienna, Austria; 2Programme for Proteomics, Paracelsus Private Medical University, 5020 Salzburg, Austria; 3Botany and Microbiology Department, College of Science, King Saud University, P.O. Box 2455, Riyadh 11451, Saudi Arabia; 4Biomedical Sciences Program, University of Science and Technology, Zewail City of Science and Technology, Giza 12511, Egypt

**Keywords:** Enniatin B, Beauvericin, cathepsin B, cathepsin D, CA074, cathepsin L, lysosomes, caspases

## Abstract

Enniatin B (ENN B) and Beauvericin (BEA) are cyclohexadepsipeptides that can be isolated from *Fusarium* and *Beauveria bassiana*, respectively. Both compounds are cytotoxic and ionophoric. In the present study, the mechanism of cell death induced by these compounds was investigated. Epidermal carcinoma-derived cell line KB-3-1 cells were treated with different concentrations of these compounds. The extracellular secretion of cathepsin B increased in a concentration-dependent manner, and the lysosomal staining by lysotracker red was reduced upon the treatment with any of the compounds. However, the extracellular secretion of cathepsin L and cathepsin D were not affected. Inhibition of cathepsin B with specific inhibitor CA074 significantly reduced the cytotoxic effect of both compounds, while inhibition of cathepsin D or cathepsin L did not influence the cytotoxic activities of both compounds. In vitro labelling of lysosomal cysteine cathepsins with Ethyl (2S, 3S)-epoxysuccinate-Leu-Tyr-Acp-Lys (Biotin)-NH2 (DCG04) was not affected in case of cathepsin L upon the treatment with both compounds, while it was significantly reduced in case of cathepsin B. In conclusion, ENN B and BEA increase lysosomal Ph, which inhibits delivery of cathepsin B from Golgi to lysosomes, thereby inducing cathepsin B release in cytosol, which activates caspases and hence the apoptotic pathway.

## 1. Introduction

ENN B and BEA belong to a class of peptides named cyclohexadepsipeptides, which contain six monomers [1]. These compounds were isolated from fungi; ENN B was isolated from *Fusarium*, while BEA was isolated from *Beauveria bassiana* [2]. These cyclohexadepsipeptides were considered as food contaminants in cereals and cereal-based products [3]. Moreover, both compounds have the ability to penetrate different layers of skin barriers to reach the epidermis and dermis, which reflect the potential risk of the contaminated food [4].

Previous studies were carried out to assess the risk of inhalation of both compounds, which confirmed no acute affects [5]. Different studies showed that both compounds exert potential cytotoxic properties due to apoptosis induction [6]. Although ENN B and BEA have the ionophoric activity to induce pores in plasma membrane, the exact molecular mechanism of cytotoxicity still unclear [7]. Several cellular changes indicate that the programmed cell death was associated with ENN B treatment. These changes included chromatin condensation, DNA fragmentation, cell shrinkage, and apoptotic body formation. The cytotoxic investigations of ENN B showed significant blocking in DNA synthesis compared to p53 knock out cells [8]. The effect of cytotoxicity of ENN B on gene expression profiling revealed that ENN B can alter the cellular energy metabolism in 3T3 and HepG2 cells [9]. On the other hand, cytotoxic investigation of BEA revealed that it extensively inhibits the cellular growth, clonogenicity, invasion and migration of A375SM human melanoma cells via promotion of caspase-dependent pathway of apoptosis through upregulation of cell death receptors, anti-, and pro-apoptotic Bcl2 proteins family members [10]. Interestingly, both ENN B and BEA have been proven to bind calmodulin, a calcium-modulated protein, and inhibit 3,5 cyclic nucleotide phosphodiesterase (PDE), an important enzyme for cell proliferation [11].

Several investigations reported the antibacterial, antiviral and pesticidal properties of both compounds. ENN B for instance showed antibacterial effects against different bacteria species, in particular, *Escherichia coli, Listeria monocytogenes, Eubacterium biforme, Enterococcus faecium, Clostridium perfringens, Bifidobacterium adolescentis, Bacillus cereus,* and *B. pumilis* [12]. Another study elucidated the insecticidal properties of ENN B against many insect species, particularly *Hemiberlesia rapax, Calliphora erythrocephala,* and *Aedes aegypti*. BEA exhibits antibacterial activities against both pathogenic Gram-positive and Gram-negative bacteria and shows an effective capacity to inhibit human immunodeficiency virus type-1 integrase [13].

ENN B increases the lysosomal permeability through lysosomal membrane protein LAMP-2 destabilization [14]. Other studies reported that ENN B could increase the lysosomal cathepsin secretion through increasing lysosomal Ph [6,14]. Other studies showed that both ENN B and BEA have the ability to change the cell, mitochondria, and lysosome membranes integrity [15]. Cathepsins were also found to play an important role in apoptosis.

Lysosomal cathepsins are classified into three classes according to their active sites, cysteine cathepsins that include cathepsins B, H, V, W, K, L, X, C, S, F, and O [16], serine cathepsins that include cathepsins G and A, and aspartic cathepsins with their two members cathepsins D and E [16,17,18]. Most of cathepsins are involved in a wide range of physiological processes, such as the breakdown of intracellular proteins, energy metabolism, and immune responses, among many other functions [19,20,21,22]. Cathepsins are also implicated in different diseases, including cancer metastasis, neurodegenerative diseases, and inflammation [23]. Usage of lysosomotropic agents Leu-Leu-Ome induced lysosomal membrane permeabilization that led to the release of lysosomal cathepsins in cytosol, which was found to cleave and activate the proapoptotic family member Bid. On the other hand, secreted cathepsins were found to degrade the antiapoptotic family member Bcl-2, Bcl-Xl, or Mcl-1. Moreover, using the general cysteine peptidase inhibitor E64-d has significantly inhibited the apoptosis by Bid cleavage and antiapoptotic protein degradation inhibition [23]. Furthermore, in vitro studies showed that recombinant cathepsins B, L, S, K, and H could cleave Bcl-2, Bcl-Xl, Mcl-1, Bak and BimEl, while Bax cleavage was not observed [23]. Additionally, cathepsin C was found to play a vital role in the extrinsic pathway of apoptosis by activating granzyme B in natural killer cells (NKs). However, in vitro activation of natural killer cells with interleukin-2 restored the cytotoxic functions of granzyme B through the cathepsin C-independent pathway [24].

Several investigations proved the importance of acidic lysosomal environment in lysosomal cathepsins sorting [25,26,27]. There are two main receptors involved in lysosomal protein localization from Golgi to endosomal compartments. The first one is the cation-dependent mannose-6-phosphate receptor (CD-MPR), also known as the 46 Kda mannose-6-phosphate receptor (MPR46) [25], and the other one is the mannose 6-phosphate/insulin-like growth factor 2 receptor (M6p/IGF2R) [26]. The majority of soluble acid hydrolases are changed with mannose 6-phosphate (M6P) residues, allowing their popularity by M6P receptors in the Golgi complex and ensuring transport to the endosomal/lysosomal compartments. Other soluble enzymatic and non-enzymatic proteins are transported to lysosomes in a M6P-independent manner mediated by receptors, which include the lysosomal integral membrane proteins sortilin or LIMP-2 [27]. Once the lysosomal cathepsins are transported by one of the M6P receptors from Golgi to lysosomal compartments, the lysosomal peptidases will dissociate from their M6P ligand while the M6P receptors shuttle between the membranes and return to Golgi compartment to sort newly synthesized lysosomal cathepsins to lysosomes [28]. Treating HepG2 cells with lysosomotropic amines NH_4_Cl was found to interfere with M6P receptors’ function by increasing the lysosomal Ph, leading to the inhibition of ligand/lysosomal hydrolases dissociation and release of newly synthesized lysosomal hydrolases from Golgi to cytosol instead of transporting them to endosomal/lysosomal compartments [29].

As both Enniatin B and Beauvericin exert cytotoxic activities by inducing mitochondrial modifications and cell cycle disruption, finally resulting in apoptotic cell death, recent studies reported a potential anticancer activity [30,31]. The objective of this study is to have a deep insight into the molecular pathways of cytotoxicity of *Fusarium* mycotoxin (ENN B) and *Beauveria bassiana* mycotoxin (BEA) and the role of some lysosomal peptidases in the apoptotic pathway of both compounds.

This study intends to fill the gap of the possible cytotoxicity mechanisms of the two mycotoxins (ENN B and BEA), potentially classified as anticancer agents [32,33]. Previous studies showed that both compounds can induce impairment of lysosomes [15,34]. In addition, we aimed at investigating if the lysosomal peptidases can contribute to the cytotoxicity of these compounds and which of them can be implicated.

## 2. Results

### 2.1. ENN B and BEA Inhibited Lysosomal Staining by Lysotracker Red Dye

It was hypothesized that both ENN B and BEA could interrupt lysosomal compartments’ Ph environment to interfere with M6P-tagged peptidases transfer from Golgi to lysosomes, which leads to peptidase release into cytosol and hence extracellular secretion induction rather than lysosomal translocation. In order to confirm this hypothesis, KB-3-1 cells were treated with ascending concentrations of ENN B or BEA prior to incubation with acidic compartment-specific dye lysotracker red, which is considered to be an excellent reagent to detect lysosomal Ph shifts [35,36]. The lysosomotropic ammonium chloride was used as positive control. Inhibition of lysosomal compartments staining by ENN B and BEA were observed (Figure 1).

According to these results, lower concentration of ENN B 0.652 μM inhibited the lysosomal compartment staining to 36%, while the same concentration of BEA inhibited the lysosomal staining to 44%. The higher the concentrations of either ENN B or BEA, the more inhibition of the lysosomal compartment staining.

### 2.2. In Vitro Labelling and Pull-Down of Cysteine Cathepsins

The approachability of the active sites of cysteine cathepsins to the activity-based probe DCG04 was investigated. DCG04 was initially intended to be a selective activity-based probe of cysteine peptidases [37]. In this experiment, the cultured KB-3-1 cells were treated with 10 μM DCG04 and/or 2.5 μM ENN B for 48 h prior to Western blot analysis of cellular extracts with streptavidin–horseradish peroxidase. The unlabeled DCG04 (E64) and/or ENN B were used as negative controls.

Two specific bands were clearly appeared in DCG04 labelled cells—the higher band was approximately 30 KDa and the lower band was approximately 26 KDa. These findings showed that the cellular contents of the upper band were clearly reduced by ENN B treatment, while they were not affected by ENN B treatment in the lower band (Figure 2).

In order to identify both bands, a pull-down experiment using streptavidin–Sepharose beads was performed, the proteins that conjugated to streptavidin–Sepharose beads were subjected to protein SDS-PAGE and Western blotting with antibodies against both cathepsin B and cathepsin L (Figure 3).

The results of pull-down experiments supported the previous results of in vitro labelling of cysteine cathepsins with DCG04, and the cellular contents of cathepsins B were reduced to 57% of ENN B non-treated cells, and this range is similar to results of in vitro labelling experiments.

On the other side, the cellular level of cathepsin L was not affected by ENN B treatment (Figure 4).

Similar results were observed with BEA—concisely, the cellular contents of cathepsins B were reduced in BEA treated cells in comparison to BEA non-treated cells, while the cellular content of cathepsin L did not show any remarkable change upon treatment with BEA (Figure 5).

In order to identify these two bands, pull-down experiments were carried out to cells treated with only DCG04 or BEA and DCG04, and protein samples were subjected to electrophoresis and immunoblotting using antibodies specific to cathepsin B and cathepsin L (Figure 6).

From those experiments, we can assume that both ENN B and BEA behave similarly, either by inhibiting the proteolytic activity of cathepsin B or by interfering with biosynthetic pathway and targeting cathepsin B from Golgi to the lysosomal compartments.

### 2.3. Extracellular Secretion Studies after Treatment with ENN B or BEA

In vitro labelling studies of cysteine cathepsins have brought two hypotheses, that cathepsin B could be inhibited by ENN B or BEA or the extracellular secretion of cathepsin B was just increased upon treatment with one of those two compounds. Extracellular secretion studies were needed to answer this question. KB-3-1 cells were treated for 24 h with different concentrations of ENN B or BEA. Media of different treatments were collected, and cell lysates were prepared from cells. Both of cell lysates and media were subjected to SDS-PAGE and Western blotting using antibodies against cathepsin B, cathepsin D, and cathepsin L. Interestingly, the cellular contents of cathepsin B were significantly decreased upon treatment with ENN B or BEA. Moreover, the situation was opposite in media. However, the cellular contents or media of cathepsin D and cathepsin L were not changed upon the treatment with any of both compounds (Figure 7).

In addition, the extracellular secretion of cathepsin B was correlated with concentrations of ENN B or BEA. According to those results, the cellular content of cathepsin B was decreased to 38% after treatment with 2.5 µM ENN B, and the expression of the same protein was not detectable after treatment with 5 µM ENN B. Examination of this secretion level of cathepsin B in medium was increased upon treatment with BEA. Therefore, the secretion levels were concentration-dependent. In summary, the extracellular secretion increased to 1.8-fold after treatment with 2.5 µM ENN B and almost two-fold after treatment with 5 µM BEA.

The same experiments were carried out with BEA, and similar results to ENN B were obtained (Figure 8). However, BEA showed differences in comparison to ENN B. The extracellular secretion of cathepsin B for instance was approximately 100% after treatment with 5 µM ENN B, while, in case of BEA, the secretion was increased to 65%. The extracellular secretion of both cathepsin L and cathepsin D did not show significant differences after treatment by different concentrations of BEA (Figure 8). These data indicate that both BEA and ENN B enhance the extracellular secretion of cathepsin B, but not cathepsin D or cathepsin L.

### 2.4. Enzymatic Assay of Cathepsins B and L after Treatment with ENN B or BEA

The target of these experiments was to study if the change in cathepsin B cellular contents was due to the change in cathepsin B expression and extracellular secretion. KB-3-1 cells were treated for 24, 48, and 72 h by different concentrations of ENN B or BEA. Cellular extracts were prepared, and media were collected. Cathepsin B enzymatic assays were performed using Fluorogenic Peptide Substrate Z-R-R-AMC. Cathepsin B activities were found to be decreased in cellular extracts after treatment by ENN B or BEA, and the inhibition was correlated with time and concentrations of both compounds (Figure 9). Treatment by 1.25 µM for 24 h resulted in 32% inhibition of cathepsin B activity, while the reduction reached 30% with the same concentration of BEA. On the other hand, two- and three-fold increasing concentration of any of both compounds resulted in more reduction of cathepsin B activities in cellular extracts. For instance, cathepsin B activities in cellular extracts were reduced more than 80% after treatment by 5 µM ENN B and about 75% after treatment by 5 µM BEA.

Interestingly, reduction of cathepsin B activities in cellular extracts were correlated with increasing of cathepsin B activities in media (Figure 10).

These data indicate that both ENN B and BEA enhancing cathepsin B extracellular secretion. Furthermore, they do not have inhibition activity on cathepsin B since the reduction of cathepsin B activities in cells was correlated with increasing of enzymatic activities in media. However, measurement of cathepsin L activities in both cellular and media using Fluorogenic Peptide Substrate Z-F-R-AMC did not show significant changes in cathepsin L activities extracts (Appendix A). In the cathepsin L activity assay, cathepsin B activity was inhibited using 10 µM cathepsin B specific inhibitor CA074. These results support the previous observation that ENN B or BEA can increase only cathepsin B extracellular secretion.

### 2.5. ENN B and BEA Inhibit the Delivery of Cathepsin B from Golgi to Lysosomes by Interfering with the Biosynthetic Pathway of This Enzyme

In this experiment, KB-3-1 cells were treated with 330 nM sortilin inhibitor, AF38469, and no significant extracellular secretion pattern changes of cathepsin B were observed, while cathepsin D and cathepsin L secretion was increased more than 50% (Figure 11), which indicated that sortilin does not play important role in cathepsin B sorting, while it plays a crucial role in cathepsin D and cathepsin L sorting. Cell treatment by sortilin inhibitor and ENN B or BEA increased secretion of cathepsin D and cathepsin L more than 85%. These results indicate that both ENN B and BEA can inhibit cathepsin B, but not cathepsin D or cathepsin L sorting by M6PR-dependent pathway inhibition. Cathepsin B can only be transported to lysosomal compartments by M6PR pathway, and apparently there is no alternative pathway. Despite this, cathepsin D and cathepsin L can rely on both pathways in which sortilin can compensate the M6PR pathway. Therefore, ENN B and BEA can induce cathepsin B, but not cathepsin D or cathepsin L extracellular secretion.

### 2.6. Effect of Ammonium Chloride Treatment on the Cytotoxicity of ENN B and BEA

To study if the lysosomal cathepsins are implicated in cytotoxic effect of ENN B and BEA, KB-3-1 cells were treated with different concentrations of ENN B or BEA in the presence or absence of 20 mM ammonium chloride at different time span. The effect was not strongly obvious at 12 h, while it was marked after 24- and 48-h treatment. An amount of 85% of cells survived upon treatment with 1.25 µM at 24 h treatment with ENN B, while it reached 61% in the presence of ammonium chloride. The effect was greater after treatment with 2.5 µM ENN B at the same time span, while about 60% survived in the absence of ammonium chloride, and the survival range reached about 50% in the presence of ammonium chloride. About 33% of cells were survived after treatment with 5 µM ENN B in the absence of ammonium chloride, and 13% of cells survived in the presence of ammonium chloride. The effect was greater and more marked when the treatment was for 48 h (Figure 12A). Similar results were obtained with BEA (Figure 12B).

These results indicate the lysosomal cathepsins are implicated in the cytotoxicity of both ENN B and BEA.

### 2.7. Influence of Cathepsin B Inhibition on ENN B and BEA Cytotoxicity

To determine which of lysosomal cathepsins are involved in ENN B or BEA cytotoxicity, KB-3-1, cells were treated with different concentrations of ENN B or BEA at different time in the presence or absence of 10 µM of cathepsin B inhibitor CA074, cathepsin D inhibitor pepstatin A, or the specific cathepsin L inhibitor Z-FY (tBu)DMK (Figure 13A). Cathepsin D or cathepsin L inhibition did not influence the cytotoxicity of any of the compounds while cathepsin B inhibitor CA074 has markedly inhibited the cytotoxic effects of both compounds. For instance, treatment with 1.25 µM ENN B for 24 h killed 31% of the cells, while the cytotoxicity was greatly inhibited to 14% when the cells treated with the same concentration of ENN B was combined with 10 µM of cathepsin B inhibitor CA074. Similar results were obtained from the experiments of BEA (Figure 13B). These results indicate that cathepsin B but not cathepsin D or cathepsin L plays an important role in both compounds’ cytotoxicity.

In order to provide a direct evidence whether cathepsin B mediates the apoptotic pathway induced by ENN B or BEA, KB-3-1, cells were treated for 24 h with different concentrations of ENN B or BEA in the presence and absence of 10 µM of cathepsin D inhibitor (pepstatin A) or the same concentration of cathepsin L inhibitor (Z-FY(tBu)DMK) or cathepsin B inhibitor (CA074). Cells were harvested, and cell lysates were prepared from cells. The lysates were then subjected to SDS-PAGE and Western blotting using antibodies against the cleaved forms of some cathepsin activation dependent apoptotic proteins (Bid, -Bax and caspase 3).

The expression of these proteins was obviously elevated upon treatment with different doses of ENN B or BEA. The effect of these compounds on the expression of these proteins was significantly reduced upon treatment with the specific cathepsin B inhibitor (CA074) (Appendix A). Interestingly, the inhibition impact was observed after the treatment with different doses of ENN B or BEA. However, the expression of these proteins showed some fluctuations after cathepsin D or cathepsin L inhibition. Cleaved Bax, for instance, was elevated after inhibition of cathepsin L with some specific doses of ENN B or BEA treated cells. The expressions of Bid and caspase 3 were reduced at some specific doses of ENN B in cathepsin L inhibited cells. Cathepsin D inhibition also showed some fluctuations.

## 3. Discussion

Previous investigations showed that ENN B has ionophoric properties to induce lysosomal membrane disruption, which increases lysosomal permeability and hence the lysosomal hydrolases release and extracellular secretion. ENN B was found to destabilize the lysosomal membrane protein LAMP-2 [14]. This study showed that both ENN B and BEA behave similarly to lysosomotropic agents to elevate lysosomal pH, which increases the secretion of some M6P tagged lysosomal hydrolases. Staining of lysosomal compartments with specific lysosomal dye, lysotracker red, was inhibited upon treatment with BEA or ENN B (Figure 1). Interestingly, the extent of inhibition was dose-dependent. These results suggested that both compounds elevate the lysosomal pH, which induces cytosolic release and extracellular secretion of some lysosomal hydrolases.

It has been reported that ENN B can induce extracellular secretion of many lysosomal cathepsins as a consequence of lysosomal membrane destabilization [7]. In this study, the extracellular secretion of many lysosomal cathepsins, in particular, cathepsin B, cathepsin D, and cathepsin L, was investigated. Surprisingly, the extracellular secretion of only cathepsin B, but not cathepsin D or cathepsin L, was found to be increased upon treatment with ENN B or BEA. In vitro active site labelling studies of lysosomal cysteine cathepsins in KB-3-1 cells showed that the active site labelling of cathepsin B was inhibited as a consequence of ENN B or BEA treatment (Figure 2 and Figure 5). The same experiments did not show any obvious changes with cathepsin D or cathepsin L. Extracellular secretion studies have supported these results.

As a consequence of ENN B or BEA treatments, the protein expression in media was significantly increased for cathepsin B, but not for cathepsin L or cathepsin D (Figure 7 and Figure 8). The cellular contents of cathepsin B were reduced after treatment by ENN B or BEA, while the cellular contents of cathepsin D or cathepsin L did not show any obvious changes (Figure 7 and Figure 8). Enzymatic assay investigation confirmed our previous findings. The cathepsin B activities were reduced in cellular extracts of ENN B or BEA treated cells, while the same experiments showed elevated cathepsin B activities in media after treatment with any of both compounds. Interestingly, the change in enzymatic activities was correlated with the dose (Figure 10).

The lysosomal cathepsins have a unique pathway pattern of N-glycosylation, hence their N-glycan is mannose 6-phosphrylated, which makes them suitable for interaction with one of the two mannose 6-phsphate specific receptors, MPR46 and M6P/IGF2R. These receptors transport the lysosomal hydrolases from Golgi to the lysosomal compartments. In the lysosomal compartments, the receptors dissociate from the ligands due to the low pH of the lysosomal compartments, and the dissociated receptors return to Golgi to bring the newly synthesized lysosomal hydrolases [27,38]. For that reason, the sorting of the lysosomal hydrolases can be easily affected by lysosomal pH changes. Our results showed that the extracellular secretion of cathepsin B was induced after ENN B or BEA treatment. However, why only cathepsin B, but not other cathepsins? Some investigations revealed another lysosomal receptor called sortilin.

Sortilin is not specific for M6p tagged proteins. Therefore, it is not pH-dependent. Interestingly, it was observed that sortilin can transport some cathepsins, such as cathepsin L, cathepsin H, and cathepsin D, but not cathepsin B [27]. In this study, cells treated with BEA or ENN B induced cathepsin B extracellular secretion. The secretion of cathepsin B was not affected after sortilin inhibitor treatment. On the other hand, cathepsin D and cathepsin L extracellular secretion were not influenced by BEA or ENN B, while treatment with one of both compounds and sortilin inhibitor significantly induced cathepsin D and cathepsin L secretion (Figure 11).

Cathepsin B inhibition, but not cathepsin D or cathepsin L, significantly reduced the cytotoxic effects of ENN B or BEA (Figure 13). Moreover, it significantly reduced the expressions of tested apoptotic proteins (Bid, Bax, and caspase 3) in ENN B or BEA treated cells (Appendix A). These data indicate that cathepsin B mediates the apoptotic pathway of both ENN B and BEA. On the other hand, the expressions of these apoptotic proteins fluctuated after inhibition of cathepsin D or cathepsin L in ENN B or BEA treated cells. 

We expect that this fluctuation was due to the presence of both cathepsin D and cathepsin L in the cytosol and their physiological roles in proteins processing and turning over. Cytosolic cathepsin D and L, for instance, play an important role in activation of some proteins that are involved in apoptosis.

In addition, cytosolic cathepsin D and L are implicated in Bid degradation and activation [39]. Cathepsin L was found to participate in caspase 3 processing [40]. Moreover, there is some extent of interaction between cathepsin D and cathepsin L. Cathepsin L, for instance, showed that it plays an important role in cathepsin D processing and turnover [41,42].

In summary, both ENN B and BEA have different mechanisms for apoptosis. The lysosomal pathway of apoptosis is involved in these mechanisms. Both compounds can induce cytosolic release and extracellular secretion of cathepsin B, but not cathepsin L or cathepsin D, to induce apoptosis. Only cathepsin B secretion was affected upon treatment with one of both compounds due to different sorting mechanisms of lysosomal hydrolases, and cathepsin B sorting has a unique M6P-dependent pathway, while cathepsin D and cathepsin L can be sorted with M6P-dependent pathway and non M6P-dependent pathway via the sortilin receptor.

In conclusion, ENN B and BEA have the ability to induce lysosomal permeabilization [7,14], and, according to our findings, ENN B and BEA can induce cathepsin B release in the cytosol and secretion in the extracellular milieu. The released cathepsin B promotes caspase-dependent apoptotic cell death.

### Future Study

Taking together our findings and other studies to future investigations, for instance, drug delivery-based nanoparticles studies on both ENN B and BEA to target certain tumor types in mice models might be developed. Therefore, we assume that this study can contribute to further therapeutic cancer research insights.

## 4. Materials and Methods

### 4.1. Cell Culture

Epidermal carcinoma-derived cell line KB-3-1 (given by Dr Shen, Bethesda, MD, USA) was used in the study. The cell line was propagated in Dulbecco’s Eagle’s medium (DMEM, Gibco, ThermoFisher Scientific, Waltham, MA, USA), supplemented with 4 mM glutamine. Antibiotics 100 units/mL penicillin and 100 ug/mL streptomycin (ThermoFisher Scientific, Waltham, MA, USA) were added, and cells were incubated at 37 °C in a 5% CO_2_ atmosphere.

### 4.2. Cytotoxicity Test

Cells were seeded in 96 well plates at a density of 2000 cells/well in 100 μL/well medium and incubated for 24 h. After that, cells were incubated separately with ENN B and BEA (Biomol, Hamburg, Germany), and concentrations ranged from 1.25 to 5 μM. The percentage of viable cells was determined at 24, 48, and 72 h post ENN B and BEA treatment by 3-(4,5-dimethylthiazol-2-yl)-2 and -5 diphenyltetrazolium bromide (MTT)-based viability assay (EZ4U, Biomedica, Vienna, Austria). An amount of 20 μL of EZ4U solution was added to each well. After incubation for 2 h at 37 °C, the absorbance was measured by a microplate reader (Infinite F200, Tecan, Männedorf, Switzerland) at 450 nm with 620 nm as reference to the unspecific background values. All experimental work was repeated three times in triplicate.

### 4.3. Peptidase Inhibitors Treatment

Cell lines (~10^7^) were treated for 24, 48, and 72 h at 37 °C in a complete medium containing 10 μM inhibitor of E64-d (Sigma-Aldrich, St. Louis, MO, USA), CA074 (Merck, Rockville, MD, USA), leupeptin (Sigma Aldrich, St. Louis, MO, USA), and pepstatin A (ThermoFischer, Waltham, MA, USA). Dimethylsulphoxide solvent (final concentration 0.1%) was used as a control.

### 4.4. Lysotracker Staining

These experiments were performed according to [24]. Cells were grown on coverslips to reach 1 × 10^6^ cells/mL. Cells were then treated separately with ENN B and BEA inhibitors for at least 24 h, and medium aspirated cells were washed twice with phosphate buffer saline (PBS) before being treated with 600 nM LysoTracker dye and DAPI staining (ThermoFischer Scientific, Dreieich, Germany) for 10 min at room temperature (RT).

Images were captured using an Olympus BX51 upright microscope with a 60× 1.4NA objective and Picture framer software (Olympus, Tokyo, Japan).

The lysosomal area was calculated using Image J software.

### 4.5. Cysteine Cathepsin Activity Assay

The cysteine cathepsin activity of cellular extract and media was determined spectrofluorimetrically according to [43]. Briefly, 2 μg of cellular protein extract were added to activation buffer (100 mM sodium acetate buffer (pH 5.5) containing 2 mM L-cysteine, 1 mM EDTA, and 0.1% (*w/v*) triton X-100). The samples were incubated for 5 min at 37 °C before the addition of 10 μM Z-F-R-AMC (Z-Phe-Arg-7-amido-4-methycoumarin) (Sigma Aldrich, Darmstadt, Germany) as substrate, then cathepsin B activity was measured using specific substrate that is hardly cleaved by another cathepsin [44], Z-R-R-AMC (Sigma Aldrich, Darmstadt, Germany). The reaction was stopped after 30 min of incubation at 37 °C with stopping solution (0.1 M chloroacetic acid, 0.1 M sodium acetate, pH 4.3). The released 7-amido-4-methycoumarin (AMC, excitation 370 nm, emission 460 nm) was measured by Infinite F200, Tecan Männedorf, Switzerland. All measurements were in triplicate.

### 4.6. Active Site Labelling of Cysteine Cathepsins

Cysteine cathepsins can be labelled in cultured cells by using the activity-based probe DCG04 (Medkoo, Morrisville, NC, USA), which is a biotinylated form of general cysteine peptidase inhibitor E-64. DCG04 is a selective activity-based probe of cysteine peptidases. It is a biotinylated form of the general cysteine peptidase inhibitor E64 (which line number). It has the ability to bind to active cysteine peptidases in protein mixture [37]. Cells were incubated for 72 h at 37 °C with 10 μM DCG04. Protein extract was then prepared. 30 μg of the extract were separated using 12.5% SDS polyacrylamide gels. Proteins were then transferred to nitrocellulose membrane (Santa Cruz biotechnology, Dallas, TX, USA). Membranes were then blocked using 3% bovine serum albumin (BSA) (ThermoFischer Scientific, Waltham, MA, USA) in PBS. Membrane was then incubated in streptavidin-horseraddish peroxidase (0.125 μg/mL PBST (BioLegend, San Diego, CA, USA) prior to enhanced chemiluminescence detection.

### 4.7. Pull-Down of DCG04 Labelled Cysteine Cathepsins

In order to pull down the in vitro labelled cysteine cathepsins, 250 μL (about 400 μg) of protein concentrations of cells extracts were measured according to Bradford 1976 [45]), and cellular extracts that were previously labelled with DCG04 were diluted with 750 μL binding buffer (20 mM sodium acetate, pH 5.5, 150 mM sodium chloride, 0.1% triton X-100, 10 μg/mL E-64, 10 μg/mL leupeptin (Sigma Aldrich, St. Louis, MO, USA), 1 mM PMSF (Abcam)), and the mixture was centrifuged at 14,000× *g* for five min, and supernatant was incubated overnight with 40 μL settled streptavidin beads at 4 °C. Beads were precipitated by centrifugation for 5 min at 3000 rpm, then they were washed five times with 20 mM sodium acetate (pH 5.5), 150 mM sodium chloride, and 0.1% triton X-100 and twice with 10 mM Tris-HCL, pH 6.8 [46]. Settled beads were mixed with 40 μL 2X sample buffer and heated five min at 95 °C [47,48,49,50]. The supernatant was subjected to SDS-PAGE and blotting on nitrocellulose membrane and immunoblotted with antibodies (1:2000) (ThermoFischer Scientific, Waltham, MA, USA) against cathepsin B, D, and L [51].

### 4.8. Lysosomal Cathepsin Inhibition

Different general cathepsin inhibitors were used to assess the effect of the lysosomal cathepsins on ENN B and BEA cytotoxicity. Around ≅ 80% confluent cells were treated with 10 μM of specific cathepsin B inhibitor CA074 [51,52]. For general serine and cysteine cathepsin, inhibition leupeptin was used at 10 μM concentration [53,54]. For cathepsin L inhibition, Z-FY(tBu)DMK was also used at 10 μM concentration [54].

### 4.9. Western Blotting

Experiments were conducted as previously described [33,34,35,36]. Initially, cells were grown in 100 mM cell culture dishes in 5% CO_2_ incubator in DMEM medium supplemented with 5% fetal bovine serum. Medium was then aspirated, and cells were washed twice with PBS. Cells were scrapped in lysis buffer (200 mM sodium acetate, 150 mM NaCL, pH 5.5 supplemented with 40 μM E-64) and transferred to 1.5 μL centrifuge tube. The cells were then homogenized on ice by ultrasonication, and then 0.1% Triton X-100 was added. The homogenized cells were then extracted on ice for 30 min. Samples were then cleared by centrifugation 15,000× *g* for 10 min. The proteins were separated under reducing conditions by 12.5% SDS-PAGE. Proteins were then transferred to nitrocellulose membrane (Santa Cruz Biotechnology, Dallas, TX, USA) by semi-dry blotting at 25 V for 30 min. Unspecific sites were then blocked for 3 h by blocking solution (3% BSA in PBS). Membrane was then incubated for 90 min with the primary antibodies, cathepsins B, D, or L (ThermoFisher Scientific, Waltham, MA, USA) (1:2000), cleaved Bid, Bcl-2, caspase 3 (Cell Signaling, Graz, Austria) (1:1000), and β-actin (Sigma Aldrich, St. Louis, MO, USA) (1:3000), and then it was washed five times with PBST and incubated another 90 min with the corresponding secondary antibodies and washed three times by PBST and one time by PBS. Enhanced chemiluminescence (Amersham ECL plus Western blotting detection reagent, GE Healthcare, Vienna, Austria) was used for visualization. Membranes were exposed to X-ray films (Amersham Hyper film ECL, GE Healthcare, Vienna, Austria). Experimental films were scanned and quantified by ImageJ (NIH, Bethesda, MD, USA).

### 4.10. Secretion Studies

In order to induce lysosomal hydrolases secretion, 80% confluent cells were treated with 20 mM NH_4_Cl for 24, 48, and 72 h. Media were then collected, and cells were harvested and subjected to protein extraction prior to Western blot analysis [55,56,57]. For inhibition of sortilin receptor, 80% confluent cells were treated with 330 nM specific sortilin inhibitor AF38469 (MCE, Vienna, Austria) for 24 h. Cells were then harvested, and proteins were extracted and subjected to SDS-PAGE and immunoblotting.

### 4.11. Statistical Analysis

Statistical analysis was performed using a non-parametric *t*-test to compare between two groups, and, if there were more than two groups, one-way ANOVA or two-way ANOVA were used according to the number of independent variables. When an analysis of variance (ANOVA) F test is significant, it does not report which pairs of means are different. Post hoc tests are used to uncover specific differences between three or more group means. Dunnett’s can be used after the ANOVA has been run to identify the pairs with significant differences. Dunnett’s test (Dunnett’s multiple comparison) compares means from several experimental groups against one fixed control group mean to see if there is any difference. The Bonferroni test is a statistical test used to reduce the instance of false positive, if needed. Bonferroni designed an adjustment to prevent data from incorrectly appearing to be statistically significant. All statistical analyses were performed with GraphPad Prism (GraphPad Software), San Diego, CA, USA, as well as Microsoft Excel 365. The probability level of *p* < 0.05 was considered statistically significant. Data were presented as mean ± SE. Detailed statistical parameters for specific experiments were described in the appropriate section or T legends.

## Figures and Tables

**Figure 1 ijms-24-02030-f001:**
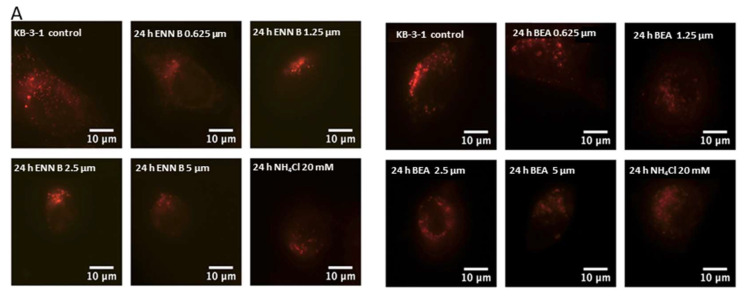
Effect of ENN B and BEA treatment on the lysosomal staining by LysoTracker red. (**A**) Fluorescence micrograph of KB-3-1 cells treated for 24 h with different concentration of ENN B or BEA prior incubation with LysoTracker red; ammonium chloride treated cells were used as a positive control. Each image represents a single cell from three different experiments. (**B**) Proportion of lysosomal area per cell after 24 h treatment ± ENN B or ammonium chloride, the area of the control cells was set to 100%. The data were analyzed using one way ANOVA with Dunnett’s post hoc analysis (*** *p* < 0.001; N = 7). Statistics was calculated using GraphPad Prism (N = samples/group and *n* = experiment replicates).

**Figure 2 ijms-24-02030-f002:**
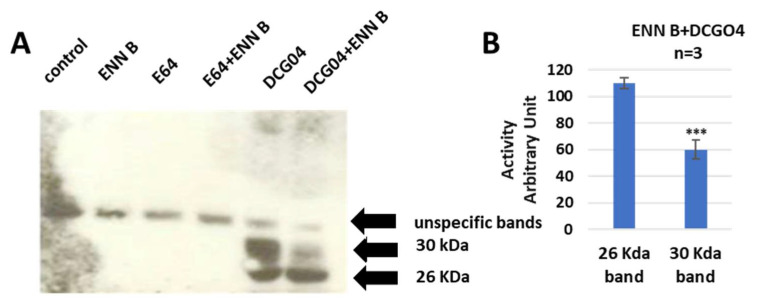
Impact of ENN B on the active site labelling of cysteine cathepsins by DCG04. (**A**) Active-site labelling with DCG04 was performed as described in materials and methods. Cells were harvested, and proteins were then subjected to protein electrophoresis and Western blotting with streptavidin–horseradish peroxidase. (**B**) Cellular content of 26 KDa and 30 KDa bands in cells treated with ENN B and DCG04 were compared to cells treated with only DCG04. The data were analyzed by *t*-test and statistics using GraphPad Prism. *** *p* < 0.001. Graphs are shown as mean ± SE, N = 7).

**Figure 3 ijms-24-02030-f003:**
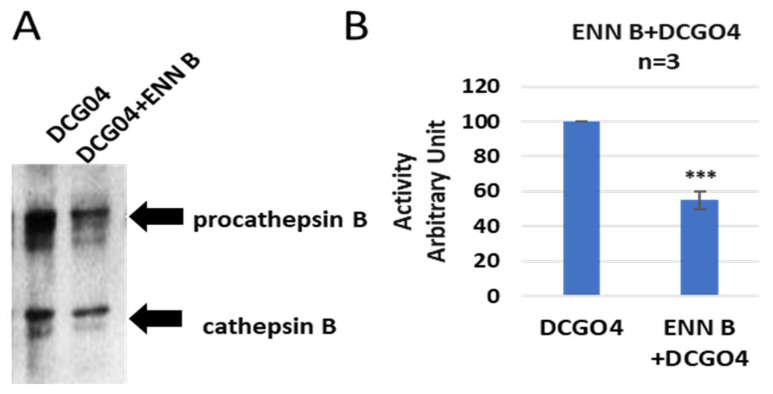
Avidin pull-down experiment of cysteine cathepsins before and after ENN B treatment. (**A**) Avidin pull-down experiment was performed as described in materials and methods. Conjugated proteins to avidin Sepharose beads were subjected to SDS-PAGE and Western blotting with antibodies specific to human cathepsin B. (**B**) Ratio of cellular contents of cathepsin B before and after treatment with ENN B. The data were analyzed by *t*-test and statistics using GraphPad Prism. *** *p* < 0.001. Graphs are shown as mean ± SE, N = 7.

**Figure 4 ijms-24-02030-f004:**
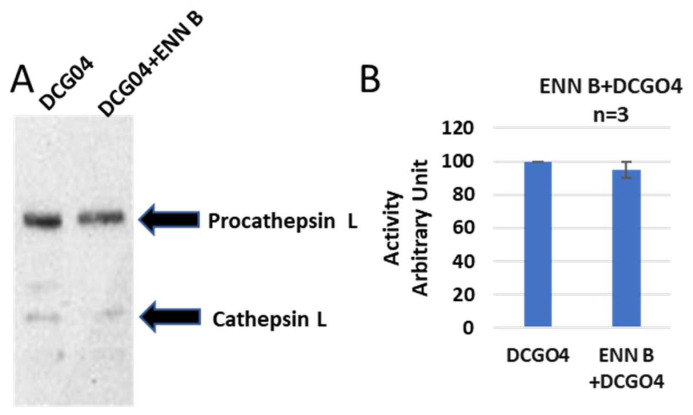
Avidin pull-down experiment of cysteine cathepsins before and after ENN B treatment. (**A**) Avidin pull-down experiment was performed as described in materials and methods. Conjugated proteins to avidin Sepharose beads were subjected to protein electrophoresis and Western blotting with antibodies specific to human cathepsin L. (**B**) Ratio of cellular contents of cathepsin B before and after treatment with ENN B. The data were analyzed by *t*-test and statistics using GraphPad Prism. Graphs are shown as mean ± SE, N = 7.

**Figure 5 ijms-24-02030-f005:**
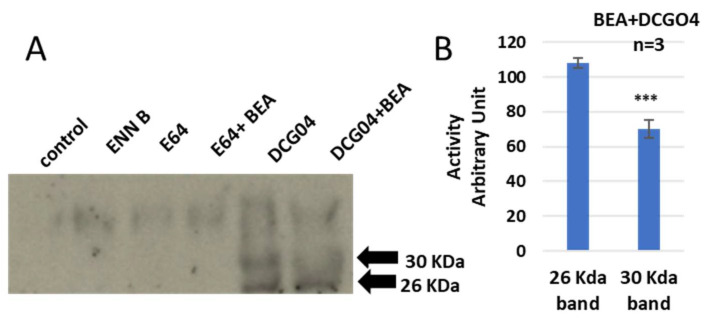
Impact of BEA on the active site labelling of cysteine cathepsins by DCG04. (**A**). Active-site labelling with DCG04 was performed as described in materials and methods. Cells were harvested and proteins were then subjected to protein electrophoresis and Western blotting with streptavidin–horseradish peroxidase. (**B**) Cellular contents of 26 KDa and 30 Kda bands in treated cells with BEA and DCG04 were compared to cells treated with only DCG04. The data were analyzed by *t*-test and statistics using GraphPad Prism. *** *p* < 0.001. Graphs are shown as mean ± SE, N = 7.

**Figure 6 ijms-24-02030-f006:**
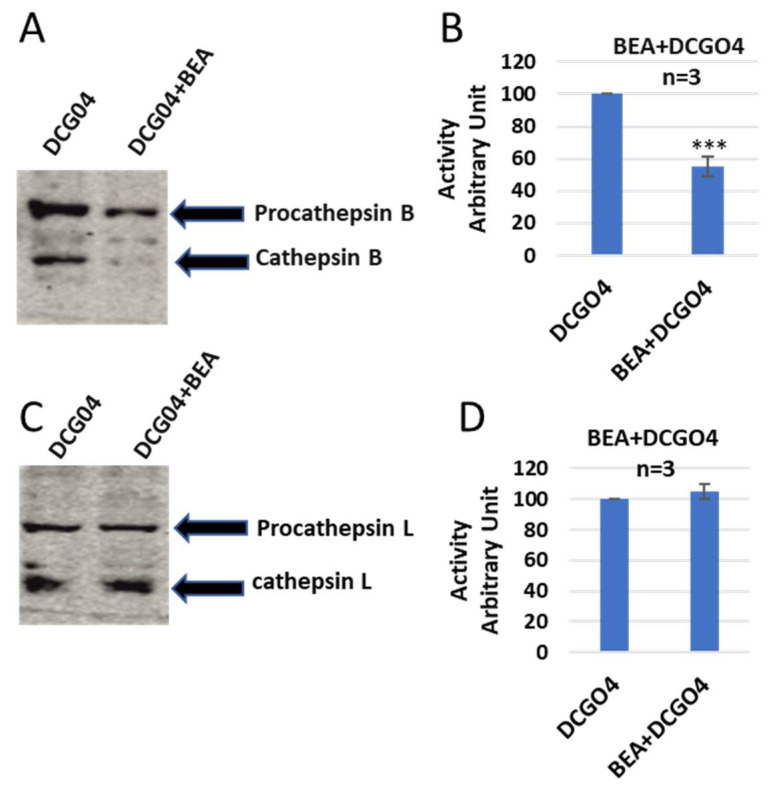
Avidin pull-down experiment of cysteine cathepsins before and after BEA treatment (**A**) Avidin pull-down experiment was performed as described in materials and methods. Conjugated proteins to avidin Sepharose beads were subjected to protein electrophoresis and Western blotting with antibodies specific to human cathepsin B. (**B**) Ratio of cellular contents of cathepsin B before and after treatment with BEA. (**C**) The protein samples were subjected to immunoblotting with anti-human cathepsin L antibodies. (**D**) Ratio of cellular contents of cathepsin L before and after treatment with BEA. The data were analyzed by *t*-test and statistics using GraphPad Prism. *** *p* < 0.001. Graphs are shown as mean ± SE, N = 7.

**Figure 7 ijms-24-02030-f007:**
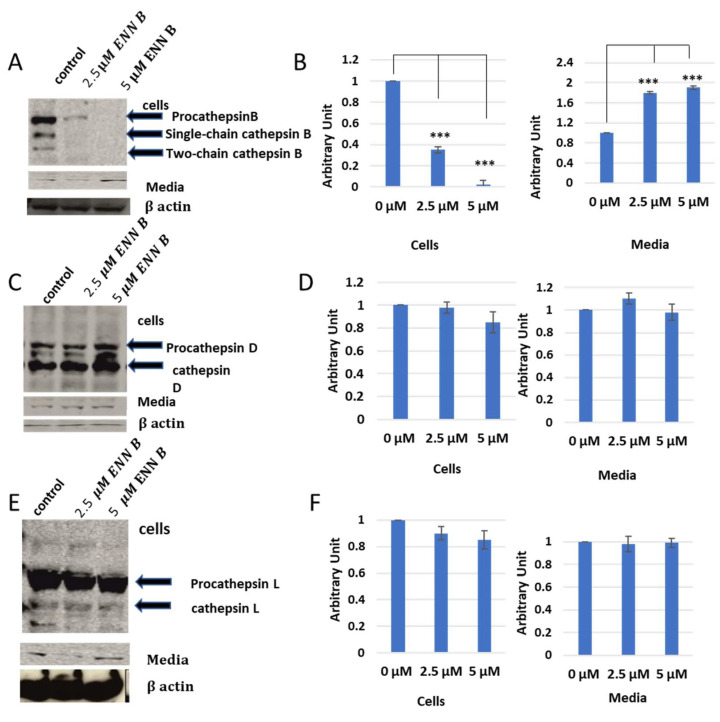
Secretion levels of cathepsin B, cathepsin D and cathepsin L after treatment with different concentrations of ENN B. (**A**) Western blot analysis of cathepsin D in cells and media. (**B**) Evaluation of cathepsin B contents in cells and media. (**C**) Cellular and media contents of cathepsin D. (**D**) Evaluation of cathepsin D in cells and media. (**E**) Cellular and media contents of cathepsin L. (**F**) Evaluation of cathepsin L contents in cells and media. The data were analyzed using one way ANOVA with Bonferroni’s and Dunnett’s post hoc analysis (*** *p* < 0.001; N = 7). Statistics were calculated using GraphPad Prism.

**Figure 8 ijms-24-02030-f008:**
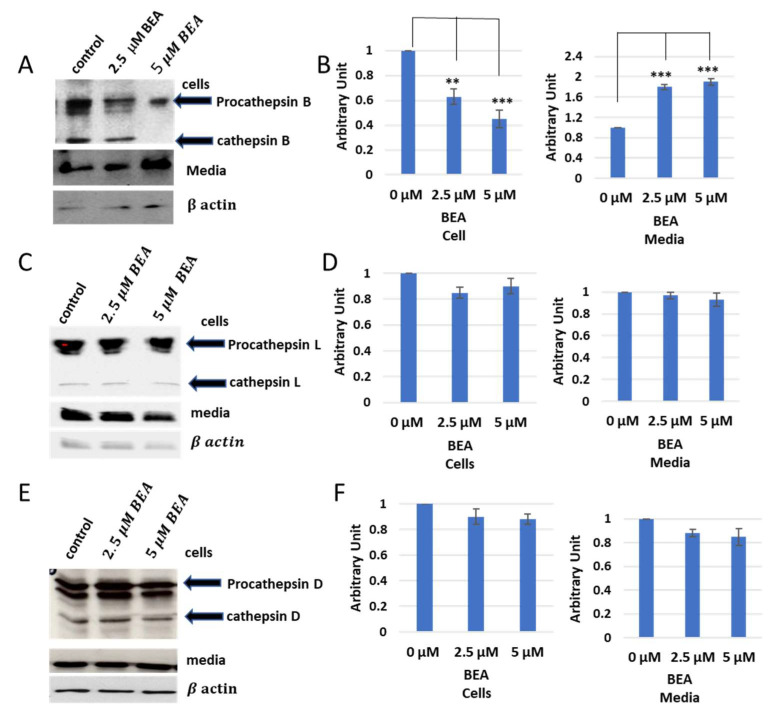
Secretion levels of cathepsin B, cathepsin D, and cathepsin L after treatment with different concentrations of BEA. (**A**) Western blot analysis of cathepsin D in cells and media. (**B**) Evaluation of cathepsin B contents in cells and media. (**C**) Cellular and media contents of cathepsin D. (**D**) Evaluation of cathepsin D in cells and media. (**E**) Cellular and media contents of cathepsin L. (**F**) Evaluation of cathepsin L contents in cells and media. The data were analyzed using one-way ANOVA with Bonferroni’s and Dunnett’s post hoc analysis (** *p* < 0.01; *** *p* < 0.001; N = 7). Statistics were calculated using GraphPad Prism.

**Figure 9 ijms-24-02030-f009:**
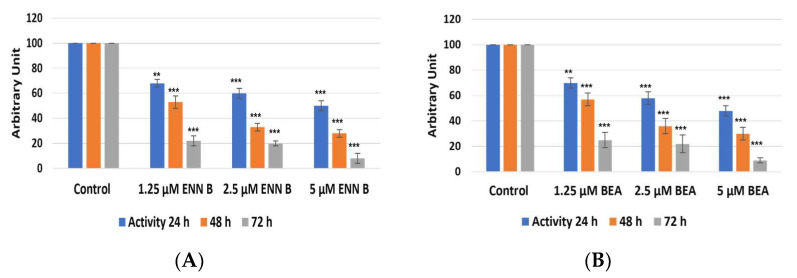
Impact of ENN B and BEA on proteolytic activities of cathepsin B in cellular extracts. (**A**) Cathepsin B activities were measured in cellular extracts using cathepsin B specific substrate Z-R-R.AMC in cellular extracts after treatment with different concentrations of ENN B at different times. (**B**) Cathepsin B activities were measured in cellular extracts after treatments with different concentrations at different times with BEA. The data were analyzed using two-way ANOVA with Dunnett’s post hoc analysis (** *p* < 0.01; *** *p* < 0.001; N = 7). Statistics were calculated using GraphPad Prism.

**Figure 10 ijms-24-02030-f010:**
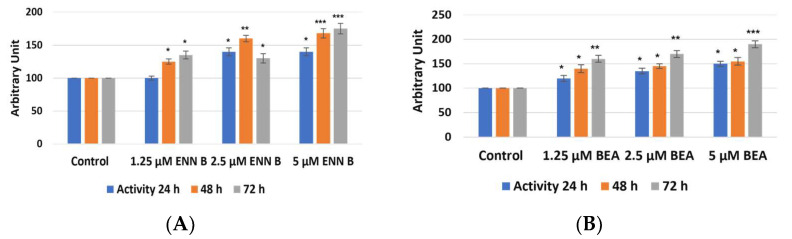
Impact of ENN B and BEA on proteolytic activities of Cathepsin B in media (extracellular). (**A**) cathepsin B activities were measured using cathepsin B specific substrate Z-R-R.AMC in media after treatment with different ENN B concentrations at different times. (**B**) Cathepsin B activities were measured in media after treatment with different concentrations of BEA at different times. The data were analyzed using two-way ANOVA with Dunnett’s post hoc analysis (* *p* < 0.05; ** *p* < 0.01; *** *p* < 0.001; N = 7). Statistics were calculated using GraphPad Prism.

**Figure 11 ijms-24-02030-f011:**
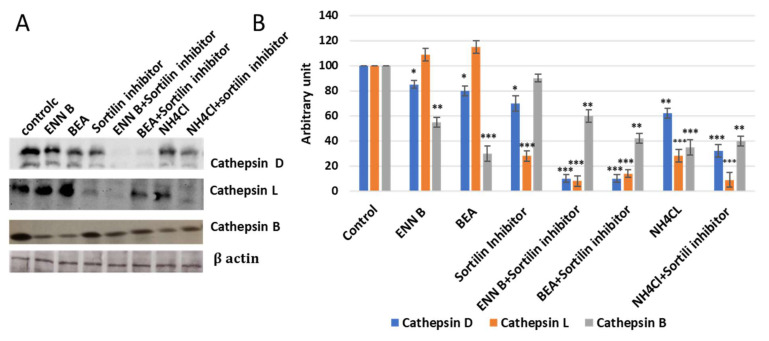
Effect of ENN B, BEA, and lysosomal receptors inhibitors on cellular retention of some lysosomal cathepsins. (**A**) Immunoblot detection of cathepsin D, cathepsin L, and cathepsin B secretion after treatment with ENN B, BEA, sortilin inhibitor, ENN B + sortilin combination, and BEA + sortilin combination. (**B**) Evaluation of cathepsin B, cathepsin D and cathepsin L secretion after treatment with ENN B, BEA, sortilin inhibitor, ENN B + sortilin combination, and BEA + sortilin combination. The data were analyzed using two-way ANOVA with Dunnett’s post hoc analysis (* *p* < 0.05; ** *p* < 0.01; *** *p* < 0.001; N = 7). Statistics were calculated using GraphPad Prism.

**Figure 12 ijms-24-02030-f012:**
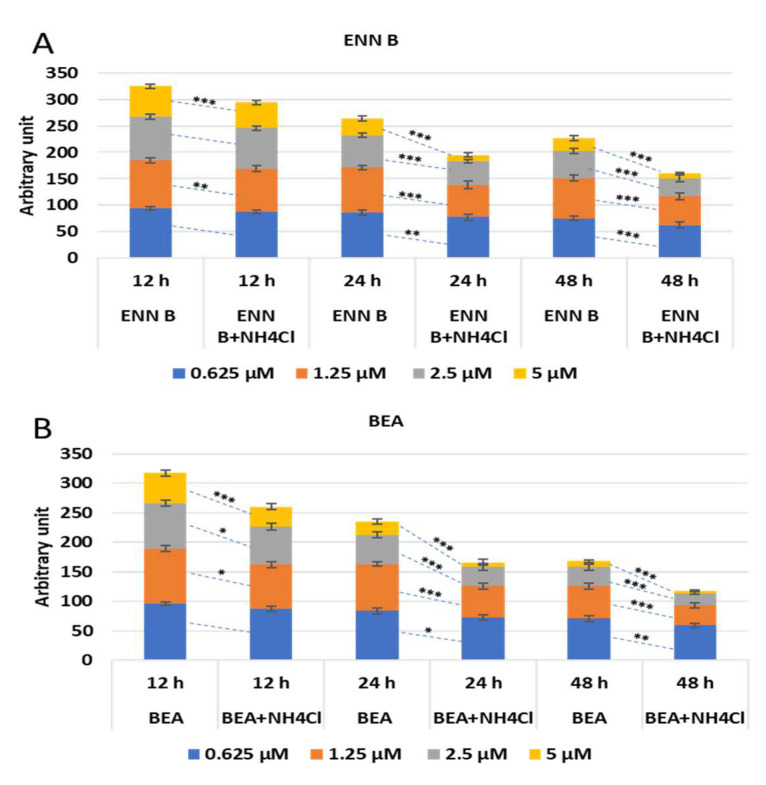
The cytotoxic effect of ENN B and BEA in the presence and absence of 20 mM ammonium chloride. (**A**) KB-3-1 cells were treated with 1.25, 2.5 and 5 µM ENN B in the presence and absence of 20 mM ammonium chloride. (**B**) KB-3-1 cells were treated with 1.25, 2.5 and 5 µM BEA in the presence and absence of 20 mM ammonium chloride. The data were analyzed using two-way ANOVA with Dunnett’s post hoc analysis (* *p* < 0.05; ** *p* < 0.01; *** *p* < 0.001; N = 7). Statistics were calculated using GraphPad Prism.

**Figure 13 ijms-24-02030-f013:**
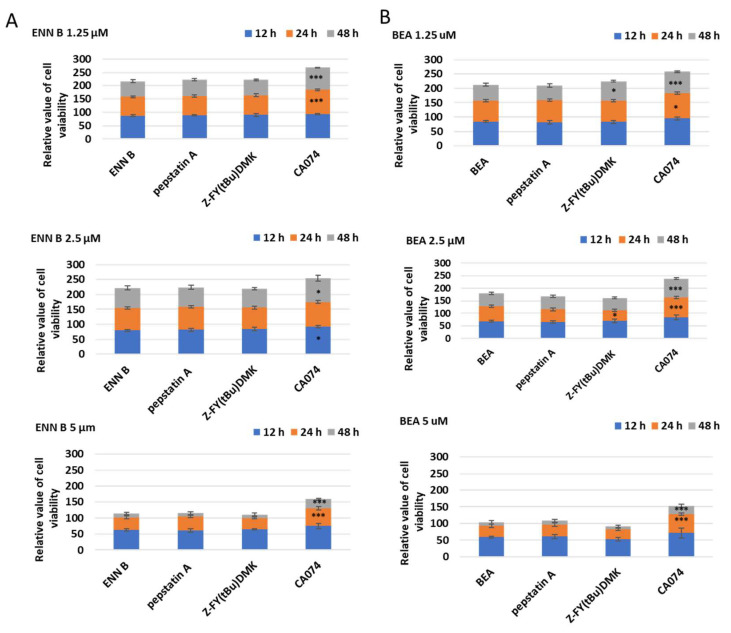
The effect of cathepsins B, D, or L inhibition on ENN B or BEA cytotoxicity. (**A**) Cells were treated with different concentrations of ENN B in the presence or absence of different cathepsin inhibitors. (**B**) Cells were treated with different concentrations of BEA in the presence or absence of cathepsins inhibitors. The data were analyzed using two-way ANOVA with Dunnett’s post hoc analysis (* *p* < 0.05; *** *p* < 0.001; N = 7). Statistics were calculated using GraphPad Prism.

## Data Availability

Data is contained within the article or Appendix A.

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
