# Peer review of "Impact of Enniatin B and Beauvericin on Lysosomal Cathepsin B Secretion and Apoptosis Induction"

_ijms, 2023, doi:10.3390/ijms24032030_

Round 1
Reviewer 1 Report
In this manuscript entitled “Impact of Enniatin B and Beauvericin on lysosomal pH which increases cathepsin B secretion and apoptosis induction”, Aufy and colleagues investigate two fungal peptides that appear to have potential health risks as contaminants in certain food products. While no acute health risks were seen through inhalation, the compounds seem to cause apoptotic cell death. Here, the authors made efforts to better understand the mechanism of cell death induction of these compounds. They found that both peptides induced cathepsin B release into the cytosol, which presumably caused caspase-mediated apoptosis. The manuscript is well written but several areas of mostly moderate concern were noted that should be addressed prior to publication of this work.
Major concerns:
- Throughout the manuscript, including the title, the authors refer to cathepsins being secreted. However, it is not particularly clear in the majority of cases weather this process refers to intracellular events (release into the cytoplasm) or release into the extracellular milieu. This distinction needs to be revisited throughout the entire manuscript and clarified on each occasion.
- Even though it is commonly accepted that cathepsins are involved in caspase cleavage and activation, which has been mentioned in Abstract and referred to in the text once (Introduction) via reference, the data to confirm these predictions were not provided in the current study. It is thus highly recommended to provide these data as a means to substantiate the claim, especially when stated in the Abstract.
Moderate/Minor concerns:
- After the Introduction: Please provide one or two sentences describing the purpose of the current study.
- 2.1. Positive control (typo)?! Also, what is the pH of 5 µM ENNB versus 20 mM NH4Cl? There is a 4000x difference needed to see similar results. How can this be explained? What is the pH of the various ENNB stock solutions relative to the very basic NH4Cl? Please summarize the results and explain briefly what the outcome of the data could mean.
- Fig. 1, adding an intermediate dose to the assay to truly show dose-response would be beneficial. As of now, lowest dose utilized in this assay already leads to a plateau.
- 2.2. Please explain in more detail what DCG-04 was used for.
- Fig. 2, if the 30 kDa band is cleaved, should this not result in an increase in the 26 kDa band? The same thought process applies to Figs. 3A, 5A and 6A. Please discuss.
- Figs. 2B, 3B, 5B and 6B, are both bands being used as the basis for the calculations?
- At the end of the Discussion: What does all this mean? Please add a concluding paragraph past the Summary statement. Translational impact? Future studies? Other conclusions?
Author Response
Dear reviewer 1
Thank you so much for reviewing our manuscript and giving us your valuable suggestions. We addressed all your concerns in the revised manuscript file. Please find the answer to your comments below.
Major concerns
Comment 1: Throughout the manuscript, including the title, the authors refer to cathepsins being secreted. However, it is not particularly clear in the majority of cases whether this process refers to intracellular events (release into the cytoplasm) or release into the extracellular milieu. This distinction needs to be revisited throughout the entire manuscript and clarified on each occasion.
Response: Thank you so much for this point and the careful reading.
Based on previous studies, Lysosomotropic agents interfere with M6P/IGF2R functions which lead to failure of newly synthesized lysosomal peptidases transport from Golgi to the lysosomal compartment leading to accumulation of those peptidases in cytosol and hence increasing of extracellular secretion of M6P receptors dependent lysosomal proteins. In addition, the lysosomotropic agents are increasing the accumulation of some lysosomal enzymes I cytosol before secretion in the extracellular milieu (Probst et al., 2013, Probst et al, 2006 and Allemailem et al., 2021)
In our study, we have checked the secretion of some lysosomal cathepsins in both cells (intracellular) and culture media (extracellular) after treatment with Enniatin B or Beauvericin, and we found that cathepsin B levels are decreased in cellular extract and increased in culture media which means that, cathepsin B is extracellularly secreted after treatment by ENN B or BEA. This was addressed in the manuscript, for example please see line 19
References:
- Probst OC, Karayel E, Schida N, Nimmerfall E, Hehenberger E, Puxbaum V,
Mach L. The mannose 6-phosphate-binding sites of M6P/IGF2R determine its
capacity to suppress matrix invasion by squamous cell carcinoma cells.
Biochem J. 2013 Apr 1;451(1):91-9. doi: 10.1042/BJ20121422. PMID:
23347038; PMCID: PMC3632087. - Probst OC, Ton P, Svoboda B, Gannon A, Schuhmann W, Wieser J, Pohlmann R, Mach L. The 46-kDa mannose 6-phosphate receptor does not depend on endosomal acidification for delivery of hydrolases to lysosomes. J Cell Sci. 2006 Dec 1;119(Pt 23):4935-43. doi: 10.1242/jcs.03283. Epub 2006 Nov 14. PMID: 17105763.
- Allemailem KS, Almatroudi A, Alrumaihi F, Almatroodi SA, Alkurbi MO, Basfar GT, Rahmani AH, Khan AA. Novel Approaches of Dysregulating Lysosome Functions in Cancer Cells by Specific Drugs and Its Nanoformulations: A Smart Approach of Modern Therapeutics. Int J Nanomedicine. 2021 Jul 26;16:5065-5098. doi: 10.2147/IJN.S321343. PMID: 34345172; PMCID: PMC8324981.
Comment 2: Even though it is commonly accepted that cathepsins are involved in caspase cleavage and activation, which has been mentioned in Abstract and referred to in the text once (Introduction) via reference, the data to confirm these predictions were not provided in the current study. It is thus highly recommended to provide these data as a means to substantiate the claim, especially when stated in the Abstract
Response: Very good point, thank you so much, we agree with this and have incorporated your suggestion in the manuscript by track changes mood (lines 390 to 402) as well as, by blots figures in the supplementary data.
Moderate/Minor concerns:
Comment 3: After the Introduction: Please provide one or two sentences describing the purpose of the current study.
Response: Agree. We have, accordingly modified to emphasize this point. Please see lines from 111 to 121
comment 4: 2.1. Positive control (typo)?! Also, what is the pH of 5 µM ENNB versus 20 mM NH4Cl? There is a 4000x difference needed to see similar results. How can this be explained? What is the pH of the various ENNB stock solutions relative to the very basic NH4Cl? Please summarize the results and explain briefly what the outcome of the data could mean.
Response:
- Positive control (typo)?!
Response: yes, was a typo and addressed accordingly, thank you
- Also, what is the pH of 5 µM ENNB versus 20 mM NH4Cl? There is a 4000x difference needed to see similar results. How can this be explained? What is the pH of the various ENNB stock solutions relative to the very basic NH4Cl?
Response: The 20 mM ammonium chloride was used as a kind of positive control of our experiment. Ammonium chloride is a lysosomotropic agent which known to interfere with the transport pathway of some M6P/tagged lysosomal proteins. According to previous literatures, ammonium chloride was used at range 10-50 mM concentrations to enhance the lysosomal peptidases secretion. We want to investigate if the cytotoxic doses of ENN B or BEA are exerting similar effects of lysosomotropic agents such as ammonium chloride or chloroquine by interfering with the transport pathways of some M6P-tagged lysosomal peptidases (Ufuk et al., 2015). The idea not to make exact comparison between ammonium chloride and ENN B or BEA rather than examination of cytotoxic doses of both compounds could interfere with the biosynthetic and delivery pathways of some lysosomal hydrolases to induce lysosomal peptidases secretion.
Reference:
- Ufuk A, Somers G, Houston JB, Galetin A. In Vitro Assessment of Uptake and Lysosomal Sequestration of Respiratory Drugs in Alveolar Macrophage Cell Line NR8383. Pharm Res. 2015 Dec;32(12):3937-51. doi: 10.1007/s11095-015-1753-8. Epub 2015 Jul 30. PMID: 26224396; PMCID: PMC4628094.
- Please summarize the results and explain briefly what the outcome of the data could mean.
Response: We agree with this and have incorporated your suggestion throughout the
manuscript. Please see lines from 454 to 463 (track changes)
Comment 5: Fig. 1, adding an intermediate dose to the assay to truly show dose-response would be beneficial. As of now, lowest dose utilized in this assay already leads to a plateau.
Response: yes, it is a good point. However, we used the doses that have shown even a minimal cytotoxic effect (0.625 mM, 72 hours), and we did not observe any cytotoxic effect at shorter times.
Comment 6: 2.2. Please explain in more detail what DCG-04 was used for.
Response: Thank you for pointing this out, we agree and added to the manuscript, lines 521 to 524
Comment 7: Fig. 2, if the 30 kDa band is cleaved, should this not result in an increase in the 26 kDa band? The same thought process applies to Figs. 3A, 5A and 6A. Please discuss.
Response: From activity-based probes we cannot obtain direct evidence of which proteins are exactly labelled. For that reason, we have performed pull-down experiments to identify which bands have been changed upon treatment with ENN B or BEA, the proteins of those experiments were exposed to SDS-PAGE and immunoblotting with specific antibodies, and we found that cathepsin B labeling was reduced after treatment with any of both compounds. The idea of Figs 2 A and 5 A to have a clear overview if some of cysteine peptidases expression/activity was changed upon the treatment. For Figs 3A and 6A these are pull-down experiments using specific antibodies, which means the lower band is the cleaved/processed form of the upper band and usually cleavage of the upper band (immature form of the protein) results in more production of the lower band (processed protein).
Comment 8: Figs. 2B, 3B, 5B and 6B, are both bands being used as the basis for the calculations?
Response: For 2 B and 5 B Figs, each of both bands was evaluated separately, because it is active site labelling of cysteine peptidases and we do not have any evidence if both bands belong to one or more proteins. For the other figures, the bands show specific proteins because we used specific antibodies. The smaller bands (mature form) are the processed form of the bigger bands (immature proteins or latent precursors). Both bands were evaluated together.
Comment 9: At the end of the Discussion: What does all this mean? Please add a concluding paragraph past the Summary statement. Translational impact? Future studies? Other conclusions?
Response: thank you so much for highlighting this issue, agree and addressed in the manuscript. Please see lines from 461 to 472.
Reviewer 2 Report
The topic of the article by Aufy at al. looks interesting; however, there are different issues which need to be addressed before considering the manuscript suitable for publication.
The introduction provides a sufficient background but lacks the research question. A short description of what the authors are trying to test should be included.
The authors, in the first result, show the effects of different concentrations of ENN B and BEA on lysosome staining. Are these concentrations cytotoxic? The effect on cell viability should be the first result, not one of the last (figure 12) where, among other things, the concentration of 0.625 uM was not even tested. Cytotoxicity at this concentration should be evaluated. Furthermore, the title of paragraph 2.1 does not seem to me to agree with the results shown.
I have highlighted serious problems in the figures, almost all of which should be revised. There are many errors and oversights both in the figure and in the caption which very often does not match what is shown in the figure. For example, in figure 1 panel B, X-axis bar graph on the right is not correct. Ammonium chloride positive or negative control? Contradiction between main text, caption, image. How was the lysosomal area determined? Specify the post hoc test used: Dunnet’s or Bonferroni? With which microscope were the images acquired? Supplier and catalog number of the lysotracker should be added (in materials and methods). Another example, figure 4 lacks the name of samples. Please check all figures and captions carefully, there is something wrong with all of them.
The reason for which the concentration of 2.5 uM of ENN B was chosen in section 2.2 should be justified as well as the reason why in section 2.3 they test both 2.5uM and 5uM.
Please review the bibliography by adding references to works published in recent years on ENN B, BEA and lysosomes and cathepsins.
The statistical analysis should be better described in the materials and methods section, where the test used is not mentioned at all. The method is written in the legends to the figures, but it is unclear: specify, as I said in the previous comment, the post hoc used where necessary
Author Response
Reviewer 2
Dear Reviewer,
Thank you so much for your efforts and time reviewing our manuscript and giving us your
valuable suggestions which will improve our work and make it more clear to the readers. Therefore, we addressed all of your concerns in the revised manuscript file. Please see below response to your comments.
Comment 1: The introduction provides a sufficient background but lacks the research question. A short description of what the authors are trying to test should be included.
Response: thank you so much for this observation, we agree with this and have incorporated your suggestion throughout the manuscript. please see lines from 111 to 121.
Comment 2: The authors, in the first result, show the effects of different concentrations of ENN B and BEA on lysosome staining. Are these concentrations cytotoxic? The effect on cell viability should be the first result, not one of the last (figure 12) where, among other things, the concentration of 0.625 uM was not even tested. Cytotoxicity at this concentration should be evaluated. Furthermore, the title of paragraph 2.1 does not seem to me to agree with the results shown.
Response:
- In figure 1 panel B, X-axis bar graph on the right is not correct it was a typo, and it is BEA and it was corrected.
- The Concentration of 0.625 µM showed a level of cytotoxicity but it did not reach the Lc50 for that reason, we did not show the cytotoxicity result of this concentrations, although we tested even other smaller concentrations in our experimental work. We added 0.625 µM results to the figures. The other higher concentrations are cytotoxic.
- From our point of view, we wanted first to elucidate if there are any lysosomal peptidases secretions as consequence of ENN B or BEA treatment and if yes, we wanted to identify which peptidases are secreted as a result of the treatment. Secondly, which pathways of lysosomal proteins delivery have been inhibited to induce the secretion was also one of our aims. Figures 1 to 11 have focused on those issues. While in Figure 12, we tried to investigate the consequences of lysosomal cathepsins secretion on cell viability.
- The title of paragraph 2.1. has been modified to be more accurate.
Comment 3: I have highlighted serious problems in the figures, almost all of which should be revised. There are many errors and oversights both in the figure and in the caption which very often does not match what is shown in the figure. For example, in figure 1 panel B, X-axis bar graph on the right is not correct. Ammonium chloride positive or negative control? Contradiction between main text, caption, image. How was the lysosomal area determined? Specify the post hoc test used: Dunnet’s or Bonferroni? With which microscope were the images acquired? Supplier and catalog number of the lysotracker should be added (in materials and methods). Another example, figure 4 lacks the name of samples. Please check all figures and captions carefully, there is something wrong with all of them.
Response: Thank you so much for the deep read and valuable observations, all your concerns were addressed accordingly:
- In figure 1 panel B, X-axis bar graph on the right it was Typo and we have corrected it.
- Ammonium chloride positive or negative control? thank you for point this out, this was a typo, and addressed in the manuscript, Ammonium Chloride was used as a positive control,
- images were captured using an Olympus BX51 upright microscope with a 60X 1.4NA objective and Pictureframer software (Olympus,Tokyo, Japan).
- The lysosomal area was calculated using ImageJ software. It was added to materials and methods sections.
- LysoTracker™ Red DND-99 (ThermoFischer Scientific, Germany).
- Post hoc test was determined and specified whither Dunnet’s or Bonferroni in the Article.
- figures and captions were checked carefully and missing names of samples were added.
Comment 4: The reason for which the concentration of 2.5 uM of ENN B was chosen in section 2.2 should be justified as well as the reason why in section 2.3 they test both 2.5uM and 5uM.
Response: In section 2.2 it was labelling experiment with activity-based probe (DCG04) and the treatment was performed at 72 h, to get reasonable amounts of labelled peptidases (it was mentioned 24 h. In the main text and, it was typo and we corrected it). The concentration of 5 µM is highly toxic, for that reason we performed this experiment with intermediate concentrations to have enough viable cells suitable for active site labelling purposes.
In section 2.3. It is a secretion study, and it was performed at 24 h, which is suitable time to get enough cells required to perform the experiments.
Comment 5: Please review the bibliography by adding references to works published in recent years on ENN B, BEA and lysosomes and cathepsins.
Response: thank you for pointing this out, agree and updated the bibliography accordingly (references from 30 to 35).
Comment 6: The statistical analysis should be better described in the materials and methods section, where the test used is not mentioned at all. The method is written in the legends to the figures, but it is unclear: specify, as I said in the previous comment, the post hoc used where necessary.
Response: Thank you so much for this recommendation agree ad addressed in the manuscript from line 581 to 595.
Reviewer 3 Report
The manuscript titled "Impact of Enniatin B and Beauvericin on lysosomal pH which increases cathepsin B secretion and apoptosis induction" describe interesting finding of the effects of Enniatin B (ENN B) and Beauvericin (BEA) on the sorting of cathepsins, especially cathepsin B. The authors found that the secretion of cathepsin B increased upon the treatment with any of the two compounds. However, the secretion of cathepsin L and cathepsin D was not affected. The data are well presented with in-depth discussion. However, the manuscript is not well written.
Comments:
1. Some expression in the manuscript need to be modified, for example the title of 2.1 is “ENN B and BEA induce lysosomal peptidases secretion”, however there is no direct evidence of peptidase secretion in this part. The title should accurately summarize the results of the corresponding section. Please modify the subtitles of the manuscript to make them concise and clear.
2. Some pictures for WB are dirty.
3. In the last part of the introduction, please supplement the scientific question, main research methods and innovation as well as the research ideas and main finding of the study.
4. Line 69, “…….through increasing lysosomal pH [6,14] 6.”, please confirm.
5. Line 118, “….ammonium chloride was used as negative control”, while in line 123-124, it was used as positive control. please confirm.
6. Figure on right side of Fig 1B, please confirm the notes on abscissa is correct.
7. Line 138, “….10μM DCG04….” to “10 μM DCG04”. Please check other sites in the manuscript.
8. Line 205, “Bea” to “BEA”. Please check other sites in the manuscript.
9. Line 383-386, “On the other hand, cathepsin D and ….. cathepsin L secretion (Fig.9), lack of any information on the secretion of cathepsin D and cathepsin L in Fig.9.
Author Response
Reviewer 3
Dear Reviewer,
Thank you so much for your great efforts and time reviewing our manuscript and giving us your valuable suggestions and comments, which will improve our paper and make it more clear to the readers and other colleagues who might use it for their upcoming related research. Therefore, we addressed all your concerns in the revised manuscript file. Please see below response to your comments.
Comment 1: Some expression in the manuscript need to be modified, for example the title of 2.1 is “ENN B and BEA induce lysosomal peptidases secretion”, however there is no direct evidence of peptidase secretion in this part. The title should accurately summarize the results of the corresponding section. Please modify the subtitles of the manuscript to make them concise and clear.
Response: Thank you so much for pointing this out, agree, and addressed in the manuscript.
Comment 2: Some pictures for WB are dirty.
Response: Thank you for pointing this out, however, we assume that you mean pictures in Figure 2. (A) and Figure 5. (A). What we want to mention is that we used streptavidin-HRP for band visualization of these experiments. The problem of streptavidin is that it is unlike specific antibodies and interacts with any biotinylated targets which can produce more background noises to the blot. Moreover, the bovine serum albumin (BSA) which is used at different stages of western blot experiments contains some biotin contaminants which contribute to background noises.
Comment 3: In the last part of the introduction, please supplement the scientific question, main research methods and innovation as well as the research ideas and main finding of the study.
Response: Thank you so much for highlighting this important point, agree, we have addressed this issue in the manuscript. Please see lines from 111 to 121.
Comment 4: Line 69, “…….through increasing lysosomal pH [6,14] 6.”, please confirm.
Response: The last 6 was a typo and we removed it.
Comment 5: Line 118, “….ammonium chloride was used as negative control”, while in line 123-124, it was used as positive control. please confirm.
Response: thank you for highlighting this point, we confirm that it was used as a positive control (typo), and it was corrected in Line 130.
Comment 6: Figure on right side of Fig 1B, please confirm the notes on abscissa is correct.
Response: It is mM, this was Typo and we have corrected it.
Comment 7: Line 138, “….10μM DCG04….” to “10 μM DCG04”. Please check other sites in the manuscript.
Thank you, agree and addressed, It is 10 μM DCG04
Comment 8: Line 205, “Bea” to “BEA”. Please check other sites in the manuscript.
Response: It is ENN B and BEA; it was typo and addressed through the whole manuscript, Thank you.
Comment 9: Line 383-386, “On the other hand, cathepsin D and ….. cathepsin L secretion (Fig.9), lack of any information on the secretion of cathepsin D and cathepsin L in Fig.9.
Response: Thank you for pointing this out, this was a typo, should be Fig 11 not 9, and we have corrected it.
Round 2
Reviewer 1 Report
In this revised version of the manuscript entitled “Impact of Enniatin B and Beauvericin on lysosomal pH which increases cathepsin B secretion and apoptosis induction”, Aufy and colleagues failed to sufficiently address prior critiques and the provision of additional data raise more questions than aiding in improving the message of the paper.
Major concerns:
- Throughout the manuscript, including the title, the authors refer to cathepsins being secreted. However, it is not particularly clear in the majority of cases weather this process refers to intracellular events (release into the cytoplasm) or release into the extracellular milieu. This distinction still needs to be revisited throughout the entire manuscript and clarified on each occasion.
- Even though it is commonly accepted that cathepsins are involved in caspase cleavage and activation, which has been mentioned in Abstract and referred to in the text once (Introduction) via reference, the data to confirm these predictions were not provided in the current study. It is thus highly recommended to provide these data as a means to substantiate the claim, especially when stated in the Abstract.
While the authors provided new data on the cleavage of putative cathepsin targets, several aspects raise additional questions:
a) Suppl. Fig. 3 (2) – ENNB, general comment. It appears the reference for the statistical calculations between treatment conditions was based on the untreated control. This is incorrect, as the goal is to gauge the impact the various inhibitors have on drug exposure. E.g., the reference for the cathepsin inhibitors –L, -D and –B for caspase 3 activation at 0.625 µM should be ENNB-only treated conditions. If one does this, it looks like caspase 3 activation is significantly reduced by cathepsin inhibitors –L and –B. How does this outcome fit the author’s story? Please address in the main text of the MS and/or in the Discussion.
b) Suppl. Fig. 3 (2) – ENNB seems to induce not only caspase 3 activation but also Bcl-2, the latter being a pro-survival factor. How is this explained? Please address in the main text of the MS and/or in the Discussion. Furthermore, cathepsin L inhibitor seems to even enhance activation of the prosurvival molecule Bcl-2 in dose-dependent fashion, except 2.5 µM. How can this be explained? Please address in the main text of the MS and/or in the Discussion.
c) Suppl. Fig. 3 (2A-D) and 4 (2A-D). It is recommended to synchronize the y-axes from 0-450 (Fig. 3) and 0-500 (Fig. 4), respectively. This might help painting a clearer picture.
d) Suppl. Fig. 4 (2) – BEA, the same reference issue as described above in a).
- The pH concerns about the 5 µM ENNB versus 20 mM NH4Cl were not addressed. In fact, changes in treatment related pH have not been experimentally confirmed/presented, yet are being referred to as causally implicated mechanistically regarding the properties of ENNB and BEA. Again, what is the pH of the various ENNB stock solutions relative to the very basic NH4Cl and how does a potential difference of this parameter contribute to the biologic consequences the authors have observed?
- Fig. 12. The same reference issue as described above in #2a.
- Fig. 13. The same reference issue as described above in #2a.
- Discussion – The authors never showed lysosomal pH increase as claimed (lines 345-346 and 350). Please address in the main text of the MS and/or in the Discussion. Consider removing this claim from the title unless experimental data will be provided.
- What do the authors mean by stating …, the labeling of cathepsin B was inhibited….? (line 356-357).
- Where did the authors show the enzymatic activities they talk about in lines 363-367?
Moderate/Minor concerns:
1. Figs. 7 and 8. It is assumed the authors meant to label the x-axes with a unit of concentration and not length?!
2. The newly added paragraphs at the end of the Discussion are nearly unreadable (lines 461-472). Also, adding references to some of the statements would help guide readers to find the relevant literature more quickly.
Author Response
Comments and Suggestions for Authors
In this revised version of the manuscript entitled “Impact of Enniatin B and Beauvericin on lysosomal pH which increases cathepsin B secretion and apoptosis induction”, Aufy and colleagues failed to sufficiently address prior critiques and the provision of additional data raise more questions than aiding in improving the message of the paper.
Major concerns:
- Throughout the manuscript, including the title, the authors refer to cathepsins being secreted. However, it is not particularly clear in the majority of cases weather this process refers to intracellular events (release into the cytoplasm) or release into the extracellular milieu. This distinction still needs to be revisited throughout the entire manuscript and clarified on each occasion.
Response:
This was revisited and addressed through the manuscript by track change mode.
For more explanation, the lysosomotropic agents as well as the compounds ENN B and BEA, can induce lysosomal permeabilization that leads to cathepsins release into cytosol and then secretion via exocytosis into the extracellular milieu (Ivanova et al 2012, Yadati et al 2020 and Bertero et al 2020). In our experiments, we found out that, cathepsin B secretion was raised upon treatment with ENN B or BEA in cultured media and decreased in cellular extract which provides strong evidence that cathepsin B is also secreted in extracellular milieu after being released in cytosol from lysosomal or Golgi compartments.
References:
- Ivanova L, Egge-Jacobsen WM, Solhaug A, Thoen E, Fæste CK. Lysosomes as a possible target of enniatin B-induced toxicity in Caco-2 cells. Chem Res Toxicol. 2012 Aug 20;25(8):1662-74. doi: 10.1021/tx300114x. Epub 2012 Jul 19. PMID: 22731695.
- Yadati T, Houben T, Bitorina A, Shiri-Sverdlov R. The Ins and Outs of Cathepsins: Physiological Function and Role in Disease Management. Cells. 2020 Jul 13;9(7):1679. doi: 10.3390/cells9071679. PMID: 32668602; PMCID: PMC7407943.
- Bertero A, Fossati P, Tedesco DEA, Caloni F. Beauvericin and Enniatins: In Vitro Intestinal Effects. Toxins (Basel). 2020 Oct 29;12(11):686. doi: 10.3390/toxins12110686. PMID: 33138307; PMCID: PMC7693699.
- Even though it is commonly accepted that cathepsins are involved in caspase cleavage and activation, which has been mentioned in Abstract and referred to in the text once (Introduction) via reference, the data to confirm these predictions were not provided in the current study. It is thus highly recommended to provide these data as a means to substantiate the claim, especially when stated in the Abstract.
While the authors provided new data on the cleavage of putative cathepsin targets, several aspects raise additional questions:
- Fig. 3 (2) – ENNB, general comment. It appears the reference for the statistical calculations between treatment conditions was based on the untreated control. This is incorrect, as the goal is to gauge the impact the various inhibitors have on drug exposure. E.g., the reference for the cathepsin inhibitors –L, -D and –B for caspase 3 activation at 0.625 µM should be ENNB-only treated conditions. If one does this, it looks like caspase 3 activation is significantly reduced by cathepsin inhibitors –L and –B. How does this outcome fit the author’s story? Please address in the main text of the MS and/or in the Discussion.
Response: Agree, Figures were modified accordingly and addressed in the manuscript (lines 401-410 and 462-479)
- Fig. 3 (2) – ENNB seems to induce not only caspase 3 activation but also Bcl-2, the latter being a pro-survival factor. How is this explained?Please address in the main text of the MS and/or in the Discussion. Furthermore, cathepsin L inhibitor seems to even enhance activation of the prosurvival molecule Bcl-2 in dose-dependent fashion, except 2.5 µM. How can this be explained? Please address in the main text of the MS and/or in the Discussion.
Response: Thank you for highlighting this issue, there was a typo/Mistake in the AB names leaded to this misunderstanding, the correct on is Bcl-2 associated X-protein (BAX) not Bcl-2, this was modified and addressed in the manuscript (same lines as 2a)
- c) Suppl. Fig. 3 (2A-D) and 4 (2A-D). It is recommended to synchronize the y-axes from 0-450 (Fig. 3) and 0-500 (Fig. 4), respectively. This might help painting a clearer picture.
Response: Agree, this was addressed as well as the figures were modified in a clear from.
- d) Suppl. Fig. 4 (2) – BEA, the same reference issue as described above in a).
Response: Agree, this was addressed as well as the figures were modified in a clear from.
- The pH concerns about the 5 µM ENNB versus 20 mM NH4Cl were not addressed. In fact, changes in treatment related pH have not been experimentally confirmed/presented, yet are being referred to as causally implicated mechanistically regarding the properties of ENNB and BEA.
Again, what is the pH of the various ENNB stock solutions relative to the very basic NH4Cl and how does a potential difference of this parameter contribute to the biologic consequences the authors have observed?
Response:
Considering the pH concerns, we disagree and believe that the pH issue is presented and addressed in the manuscript, we have confirmed the lysosomal pH change experimentally by staining the acidic compartments via lysotracker red dye. LysoTracker Red is a cell-permeable red fluorescent dye that stains acidic compartments within the cell, such as lysosomes (new text was added for more clarity, please see lines 125-133). We believe that the inhibition of the lysosomal staining by lysotracker red dye upon treatment with ENN B or BEA provides strong evidence that both compounds are able to change the lysosomal pH.
Moreover, the pH change of lysosomes as consequence of ENN B or BEA was not only our findings, while several investigations have mentioned that, the mode of toxic action of BEA and ENNB is the disruption of ion and pH homeostasis, which in turn disturbs normal cell- and organelle function (Søderstrøm et al 2022, Kamyar et al 2004, Kouri et al 2005, Tonshin et al 2010 and Wu et al 2018).
We have found out that, the extracellular secretion of cathepsin B but not cathepsin D or L was raised upon treatment with ENN B or BEA, we assume this provides more evidences for the pH change rather than mechanical changes since the three enzymes are localized in the same compartments. Almost all lysosomal hydrolases can be transported from Golgi to lysosomes via one of two pH-dependent receptors, MPR/IGF2R and MPR46. These two receptors are highly sensitive to any pH change which might results in failure of transporting the newly synthesized hydrolases from Golgi to lysosomes, which leads to cathepsins release in cytosol rather transported. Interestingly, cathepsins D and L but not cathepsin B have an alternative non-pH-dependent receptor (sortilin) which compensates the two pH-dependent receptors. This information was mentioned in the introduction and discussion parts. (443-461)
Regarding the comparison between the positive control ammonium chloride and ENN B or BEA. First, we disagree with what you have mentioned that the ammonium chloride is a very basic, we believe it is a slightly acidic come from a strong acid and weak base, while the ENN B pH was not mentioned by the manufacturer or in the previous literatures but it has basic features since it is a Hydrogen bond acceptor (https://pubchem.ncbi.nlm.nih.gov/compound/Enniatin-B).
Moreover, we used NH4Cl because it is a lysosomotropic agent which is a good positive control candidate, and it is already known that lysosomotropic agents can induce lysosomal hydrolases secretion. We have used 20 mM concentration of NH4Cl based on the previous literatures which recommend 10-50 mM concentration range (please see below references). According to many previous investigations, there are many other lysosomotropic agents can be used at very low concentrations to induce lysosomal secretion, such as Siramesine, chloroquine, azithromycin and DPP4mT, for instance can be used at micromolar concentrations to induce similar effects to mM concentrations of NH4Cl (Geng et al 2010, Ostenfeld et al 2008 and Lovejoy et al 2010). Again, we used this positive control to show that ENN B and BEA can induce cathepsin secretion, the similar effect of the lysosomotropic agents.
References:
- Sofie Søderstrøm, Kai K Lie, Anne-Katrine Lundebye, Liv Søfteland, Beauvericin (BEA) and enniatin B (ENNB)-induced impairment of mitochondria and lysosomes - Potential sources of intracellular reactive iron triggering ferroptosis in Atlantic salmon primary hepatocytes,Food and Chemical Toxicology, Volume 161, 2022,112819, ISSN 0278-6915.
- Kamyar M, Rawnduzi P, Studenik CR, Kouri K, Lemmens-Gruber R. Investigation of the electrophysiological properties of enniatins. Arch Biochem Biophys. 2004 Sep 15;429(2):215-23. doi: 10.1016/j.abb.2004.06.013. PMID: 15313225.
- Kouri K, Duchen MR, Lemmens-Gruber R. Effects of beauvericin on the metabolic state and ionic homeostasis of ventricular myocytes of the guinea pig. Chem Res Toxicol. 2005 Nov;18(11):1661-8. doi: 10.1021/tx050096g. PMID: 16300374.
- Tonshin AA, Teplova VV, Andersson MA, Salkinoja-Salonen MS. The Fusarium mycotoxins enniatins and beauvericin cause mitochondrial dysfunction by affecting the mitochondrial volume regulation, oxidative phosphorylation and ion homeostasis. Toxicology. 2010 Sep 30;276(1):49-57. doi: 10.1016/j.tox.2010.07.001. Epub 2010 Jul 16. PMID: 20621153.
- Ying Geng, Latika Kohli, Barbara J. Klocke, Kevin A. Roth, Chloroquine-induced autophagic vacuole accumulation and cell death in glioma cells is p53 independent, Neuro-Oncology, Volume 12, Issue 5, May 2010, Pages 473–481
- Lovejoy DB, Jansson PJ, Brunk UT, Wong J, Ponka P, Richardson DR. Antitumor activity of metal-chelating compound Dp44mT is mediated by formation of a redox-active copper complex that accumulates in lysosomes. Cancer Res. 2011 Sep 1;71(17):5871-80. doi: 10.1158/0008-5472.CAN-11-1218. Epub 2011 Jul 12. PMID: 21750178.
- Fig. 12. The same reference issue as described above in #2a.
Response: Agree, this was addressed as well as the figures were modified in a clear from.
- Fig. 13. The same reference issue as described above in #2a.
Response: Agree, this was addressed as well as the figures were modified in a clear from.
- Discussion – The authors never showed lysosomal pH increase as claimed (lines 345-346 and 350). Please address in the main text of the MS and/or in the Discussion. Consider removing this claim from the title unless experimental data will be provided.
Response: This was addressed in the manuscript, please see response of comment 3 above, which explain in detail the pH issue, please also check lines 125-133 and 417-423.
Although we believe the data were provided, but we don’t see a problem to modify and improve the title for more clarity:
((Impact of Enniatin B and Beauvericin on lysosomal cathepsin B secretion and apoptosis induction))
- What do the authors mean by stating …, the labeling of cathepsin B was inhibited….? (line 356-357).
Response: Labelling experiments are describing the status of enzymatic activities rather than abundance. We used this method to investigate the level of some active cathepsins in cells before and after treatment with ENNB or BEA. We have added the term active-site labelling to be more clear. Please see lines 166, 429 and 430.
- Where did the authors show the enzymatic activities they talk about in lines 363-367?
Response: Agree, however, we think the line number is different.
This was addressed, and the figure number showing the enzymatic activities was added in the related part of the discussion (line 441).
Moderate/Minor concerns:
- 7 and 8. It is assumed the authors meant to label the x-axes with a unit of concentration and not length?!
Response: Thank you for pointing this out, yes, this is a unit of concentration, and addressed in the mentioned figures.
- The newly added paragraphs at the end of the Discussion are nearly unreadable (lines 461-472). Also, adding references to some of the statements would help guide readers to find the relevant literature more quickly.
Response: The text was updated as well as supported by references. (Lines 493-507).
Reviewer 2 Report
The authors answered satisfactorily to my requests and used them as suggestions for improving the manuscript
Author Response
Thank you very much for your comments and that our changes were satisfactory for you.
Round 3
Reviewer 1 Report
All concerns adequately addressed.